# Thymopentin Enhances Antitumor Immunity Through Thymic Rejuvenation and T Cell Functional Reprogramming

**DOI:** 10.3390/biomedicines13102494

**Published:** 2025-10-13

**Authors:** Md Amir Hossain, Ye Zhang, Li Ji, Yumei Chen, Yue Luan, Yaxuan Si, Yuqing Fang, Junlan Qiu, Zhuo Wang, Guilai Liu

**Affiliations:** 1Center for New Drug Safety Evaluation and Research, State Key Laboratory of Natural Medicines, China Pharmaceutical University, Nanjing 211198, China; 2Department of Geriatrics, Nanjing Drum Tower Hospital, The Affiliated Hospital of Nanjing University Medical School, Nanjing 210002, China; 3Department of Oncology and Hematology, Suzhou Hospital, The Affiliated Hospital of Nanjing University Medical School, Suzhou 215153, China; 4School of Pharmacy, Nanjing University of Chinese Medicine, Nanjing 210023, China

**Keywords:** thymopentin, cancer immunotherapy, T cell exhaustion, cancer

## Abstract

**Background/Objectives**: T cell dysfunction represents a fundamental barrier to effective cancer immunotherapy. Although immune checkpoint blockades and adoptive cell transfer have achieved clinical success, therapeutic resistance remains prevalent across cancer types. Thymopentin (TP5), a synthetic immunomodulatory pentapeptide (Arg-Lys-Asp-Val-Tyr), has demonstrated immunostimulatory properties, yet its anticancer potential remains unexplored. The aim of this study was to investigate TP5’s antitumor efficacy and underlying immunological mechanisms. **Methods**: We evaluated TP5’s therapeutic effects in multiple murine tumor models, including B16-F10 melanoma, MC38 colorectal carcinoma, Hepa 1-6, and LM3 hepatocellular carcinoma. Immune cell populations and functional states were characterized using flow cytometry, ELISAs, and immunofluorescence analyses. The potential of TP5 as an adjuvant for T cell-based therapies was also systematically assessed. **Results**: The TP5 treatment markedly suppressed tumor growth across caner models through strictly T cell-dependent mechanisms. Critically, TP5 promoted thymic rejuvenation under immunocompromised conditions, restoring the thymus–tumor immunological balance and revitalizing peripheral T cell immunity. TP5 functionally reprogrammed T cell states, preserving effector function while ameliorating exhaustion. Furthermore, TP5 demonstrated synergistic efficacy when combined with adoptive T cell therapies, enhancing both proliferation and effector functions. **Conclusions**: TP5 represents a promising immunomodulator that addresses fundamental limitations of current T cell therapies by simultaneously enhancing T cell function and reversing thymic involution under immunocompromised conditions. Our findings provide compelling evidence for TP5’s clinical translation in cancer treatment.

## 1. Introduction

Cancer represents the second leading cause of mortality worldwide and is projected to become the predominant cause of death in the 21st century [1]. The emergence of immuno-oncology has revolutionized cancer therapeutics, in particular T cell-based immunotherapies achieving promising clinical outcomes across diverse malignancies [2,3,4]. Notably, combinatorial approaches that pair T cell-based therapies with complementary treatment modalities to enhance T cell functionality within the tumor microenvironment (TME) have emerged as a particularly promising frontier in oncology [4]. However, the clinical impact of these therapies remains constrained by widespread therapeutic unresponsiveness, with substantial patient populations exhibiting poor responses attributed primarily to progressive T cell dysfunction within the immunosuppressive TME. Consequently, preserving and augmenting T cell effector functions constitutes a critical therapeutic imperative for improving outcomes for patients with advanced cancer.

Thymopentin (TP5), a synthetic pentapeptide comprising arginine, lysine, aspartic acid, valine, and tyrosine (corresponding to residues 32–36 of thymopoietin), retains the complete biological activity of its parent hormone in governing T cell phenotypic differentiation [5,6]. As an approved therapeutic agent in China, TP5 has demonstrated clinical efficacy across diverse immunological disorders, including rheumatoid arthritis, acquired immunodeficiency syndrome, and chronic hepatitis B infections [7,8]. Mechanistic investigations have established TP5’s pivotal role in immunoregulation and T cell maturation, highlighting its capacity to modulate immune responses [9,10]. Despite its extensive clinical application and well-characterized immunomodulatory properties, TP5’s therapeutic potential in cancer immunotherapy remains unexplored.

Here, we demonstrate that TP5 potently suppresses tumor growth across multiple murine cancer models strictly through T cell-dependent mechanisms. Critically, we observed that TP5 promotes thymic rejuvenation under immunosuppressive conditions, restoring the immunological balance between thymus and tumor environments. Our study further reveals that TP5 enhances T cell cytotoxicity by simultaneously augmenting effector cytokine production and downregulating inhibitory receptor expression. Furthermore, we demonstrate TP5’s synergistic efficacy as an adjuvant for adoptive T cell therapies, providing a novel immunomodulatory strategy to amplify therapeutic responses and overcome current limitations in cancer immunotherapy.

## 2. Materials and Methods

### 2.1. Animals

Male C57BL6 and BALB/c mice (6–8 weeks old, weighing 18–22 g) were obtained from Beijing Vital River Laboratory Animal Technology Co., Ltd. (Beijing, China) BALB/c nu/nu mice (4–6 weeks old, weighing 16–18 g) were collected from GemPharmatech Co., Ltd. (Shanghai, China), and OT-I mice (C57BL/6-Tg) were presented by Dr. Yizhi Yu from the Shanghai Institute of Immunology. All animals were housed in a specific pathogen-free animal facility and acclimatized for at least one week in their cages under specific conditions, including a 12 h/12 h light/dark cycle, and were habituated to the experimental conditions one week before the experiments.

### 2.2. Primary T Cell Isolation and Cell Culture

Splenic CD8^+^ T lymphocytes were extracted using CD8a (Ly-2) magnetic microbeads (Miltenyi Biotec, San Diego, CA, USA) via magnetic separation. Isolated cells underwent activation through concurrent stimulation with plate-bound anti-CD3 and anti-CD28 monoclonal antibodies for 48 h under standard culture conditions. Mouse B16-F10 (Shanghai Cell Collection, Shanghai, China) or B16-OVA melanoma cells (gifted by Prof. Li Yongyong, Tongji University, Shanghai, China) were maintained in RPMI-1640 (Biological Industries, Cromwell, CT, USA) supplemented with 1% antibiotics and 10% fetal bovine serum. Hepa 1-6 and HCCLM3 hepatoma cells (Shanghai Cell Collection, Shanghai, China) were cultured in DMEM (Biological Industries, USA) containing identical serum and antibiotic concentrations. MC38 cells were propagated in DMEM containing 10% FBS, 1% antibiotics, 1% HEPES solution (Sigma, St. Louis, MO, USA), 1 mM sodium pyruvate (Gibco, Carlsbad, CA, USA), 50 µg/mL gentamicin (Gibco, Carlsbad, CA, USA), and 1 mM non-essential amino acids (Gibco, Carlsbad, CA, USA). All cellular cultures were maintained at 37 °C in a humidified atmosphere containing 5% CO_2_.

### 2.3. Experimental Animal Models and Therapeutic Interventions

#### 2.3.1. Tumor Models

C57BL/6J mice received subcutaneous inoculations of B16-F10 (5 × 10^5^ cells/mouse; *n* = 5 mice per group for Figure 1E–H and *n* = 7 mice per group for Appendix A) or B16-OVA cells (5 × 10^5^ cells/mouse; *n* = 5 mice per group), Hepa 1-6 cells (2 × 10^6^ cells/mouse; *n* = 11 mice per group), or MC38 cells (1 × 10^6^ cells/mouse; *n* = 8 mice per group) in the right flank to establish melanoma, hepatocellular carcinoma, and colorectal carcinoma models, respectively. Immunodeficient BALB/c nu/nu mice were inoculated with HCCLM-3 cells (1 × 10^6^ cells per animal; *n* = 5 mice per group) to generate xenograft hepatocellular carcinoma models. Tumor dimensions were monitored biweekly using precision digital calipers, with volumes calculated using the formula V = 0.5 × length × width^2^.

#### 2.3.2. Cyclophosphamide-Induced Immunosuppressive Model

To establish an immunosuppression model, BALB/c mice were stratified by body weight into experimental cohorts (*n* = 6 mice per group) and administered cyclophosphamide (80 mg/kg) via intraperitoneal injection on days 1–3 and day 10. During the recovery phase (days 4–10), animals received daily subcutaneous administration of TP5 or vehicle control.

#### 2.3.3. Pharmacological Treatments

Animals were randomized into treatment groups based on body weight and tumor burden. Pharmaceutical-grade TP5 (99.5% purity, Hainan Zhonghe Pharmaceutical, Haikou city, China) was administered subcutaneously at 20 mg/kg daily, with control animals receiving equivalent volumes of sterile saline [11]. Where indicated, sorafenib was administered at 30 mg/kg daily to the tumor-bearing mice as a positive control.

#### 2.3.4. Adoptive T Cell Therapy and Cytokine Detection by ELISA

CD8^+^ T lymphocytes isolated from OT-1 transgenic mice underwent ex vivo activation with anti-CD3/CD28 antibodies, with experimental groups receiving concurrent TP5 supplementation (1 µg/mL). Following 48 h activation, cells were expanded in IL-2-containing medium with or without TP5 (1 µg/mL) for an additional 48 h before transfer into B16-OVA tumor-bearing recipients. For cytokine analysis, cultured CD8^+^ T cells were treated with TP5 (1 µg/mL) or vehicle control, and culture supernatants were harvested on day 5 for IFN-γ and TNF-α quantification using commercial ELISA kits (Dakewe Biotech).

### 2.4. Flow Cytometry Analysis

Splenocyte suspensions were prepared by mechanical disruption of harvested spleens in phosphate-buffered saline, followed by filtration through 70 µm cell strainers and erythrocyte lysis using commercial lysis buffer (Thermo Fisher Scientific, Waltham, MA, USA). Tumor-infiltrating lymphocytes (TILs) were isolated through enzymatic digestion using collagenase-D (2 mg/mL) and DNase-I (100 µg/mL) in RPMI medium containing 2% FBS, followed by gentleMACS™ dissociation at 37 °C for 30 min. Single-cell suspensions were obtained via 70 µm filtration and density gradient separation using 40% and 70% Percoll solutions (GE Healthcare, Chicago, IL, USA). Surface marker analysis employed fluorochrome-conjugated antibodies targeting viability markers, CD3ε, CD4, CD8, PD-1, and Tim-3. For intracellular cytokine detection, thymocytes were stimulated with ionomycin (1 µg/mL), phorbol 12-myristate 13-acetate (25 ng/mL), and brefeldin-A (10 µg/mL) for 6 h at 37 °C, followed by surface staining, fixation, permeabilization, and intracellular staining for IFN-γ, TNF-α, and IL-2. Data acquisition was performed using an Attune NxT flow cytometer with subsequent analysis via FlowJo software 10 (Tree Star, OR, USA).

### 2.5. Cell Viability Assay

The Hepa 1-6 cell line was seeded into 96-well plates (0.8 × 10^4^ cells per well), and cells were treated with different concentrations of TP5 (10 ng/mL, 100 ng/mL, and 1000 ng/mL). The TP5-treated cells were collected at 0 h, 6 h, 12 h, 24 h, 48 h, and 72 h to measure the cell viability using the Sulforhodamine-B (SRB) assay, according to the previously described method [12].

### 2.6. Statistical Analysis

Statistical analysis in this study was performed using GraphPad Prism 10 (GraphPad Software, Inc. USA). Data are presented as mean ± SEM. *p* values were calculated by a two-tailed *t*-test for comparison between two groups, and when there were more than two groups, one-way or two-way analysis of variance (ANOVA) was used. *p* < 0.05 was considered statistically significant. * *p* < 0.05, ** *p* < 0.01, and *** *p* < 0.001.

## 3. Results

### 3.1. TP5 Demonstrates Broad-Spectrum Antitumor Efficacy Across Multiple Cancer Models

To investigate TP5’s therapeutic potential in oncology, we systematically evaluated its antitumor activity across diverse murine cancer models. First, we treated Hepa1–6 hepatocellular carcinoma-bearing mice with TP5 and found that the TP5 treatment significantly inhibited Hepa 1-6 primary tumor growth in comparison to vehicle-treated controls (control vs. TP5: 666.01 ± 88.62 mm^3^ vs. 341.1 ± 61.97 mm^3^, ~48% tumor inhibition, *p* < 0.001; Figure 1A–D). The therapeutic efficacy of TP5 extended beyond hepatic malignancies, demonstrating substantial growth inhibition in both B16-F10 melanoma (Appendix A) and MC38 colorectal carcinoma models (Appendix A–F), indicating broad-spectrum antitumor activity independent of tumor types.

**Figure 1 biomedicines-13-02494-f001:**
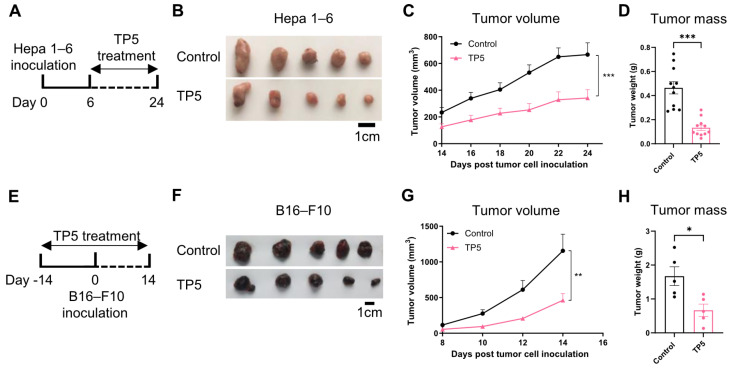
TP5 controls tumor growth in murine tumor models. (**A**–**D**) C57BL/6 mice were inoculated with Hepa1-6 cells (2 × 10^6^ cells/mouse) to generate subcutaneous hepatoma model and treated with TP5 (20 mg/kg/day) from day 6 to day 24: (**A**) experimental timeline of Hepa 1-6 model; (**B**) representative tumors from one representative experiment; (**C**) progression of Hepa 1-6 tumors; (**D**) tumor mass on day 24, data are pooled from two individual experiments, (*n* = 11 mice per group for (**A**–**D**)). (**E**–**H**), C57BL/6 mice were treated with TP5 (20 mg/kg/day) two weeks prior to B16-F10 melanoma cell (5 × 10^5^ cells/mouse) inoculation and continued for another 14 days, (**D**) experimental timeline of B16-F10 cancer prevention model, (**B**) representative tumors, (**C**) progression of B16-F10 tumors, (**D**) tumor mass on day 14 (*n* = 5 mice per group for (**E**–**H**)). *p* values were calculated using ordinary two-way analysis of variance (ANOVA) for (**C**,**G**) and unpaired two-tailed *t*-tests for (bar graphs in (**D**,**H**)). Data presented as mean ± SEM. * *p* < 0.05, ** *p* < 0.01, and *** *p* < 0.001.

To further assess TP5’s therapeutic versatility, we evaluated its cancer preventive potential using a prophylactic treatment paradigm. C57BL/6J mice received the TP5 administration for two weeks prior to the B16-F10 melanoma cell inoculation, with the treatment continuing for an additional two weeks post-challenge. This prophylactic intervention protocol demonstrated marked tumor growth suppression (control vs. TP5: 1155.11 ± 232.42 mm^3^ vs. 461.36 ± 93.85 mm^3^, ~60% tumor inhibition, *p* < 0.01; Figure 1E–H), establishing TP5’s efficacy in both therapeutic and preventive contexts. Collectively, these findings demonstrate that TP5 exhibits consistent and potent antitumor activity across multiple cancer models, suggesting a fundamental mechanism of action that transcends tumor-specific characteristics and positioning TP5 as a promising broad-spectrum anticancer therapeutic agent.

### 3.2. The Antitumor Activity of TP5 Relies on T Cell-Mediated Immunity

Given TP5’s pentapeptide composition (arginyl-lysyl-aspartyl-valyl-tyrosine) and the established roles of these constituent amino acids in T cell biology [13], we hypothesized that the effect of TP5 in controlling tumor progression operates through T cell-mediated antitumor mechanisms. To rigorously test this hypothesis, we employed athymic BALB/c nu/nu mice, which lack functional thymic development and are consequently deficient in mature T cell populations. Sorafenib was selected as the positive control because it is a clinically validated first-line standard treatment for advanced hepatocellular carcinoma (HCC), with well-documented immunomodulatory activity that is partially dependent on T cell function—making it an appropriate control for confirming the specificity of TP5’s T cell-dependent mechanism [14,15]. The immunodeficient BALB/c nu/nu mice were challenged with LM3 hepatocellular carcinoma cells and treated with TP5, sorafenib (positive control), or the vehicle. We observed that while sorafenib effectively inhibited the LM3 tumor growth compared to the vehicle control, the TP5 treatment failed to suppress the tumor progression in the T cell-deficient environment (Figure 2A–D). This striking contrast to TP5’s robust antitumor activity in immunocompetent hosts established the T cell-dependent nature of TP5’s therapeutic mechanism.

To exclude direct cytotoxic effects on malignant cells, we performed in vitro cytotoxicity assessments using the Sulforhodamine-B assay. The TP5 treatment across a broad concentration range (10–1000 ng/mL) demonstrated no detectable impact on the Hepa1–6 cell viability over extended culture periods (Figure 2E), confirming the absence of intrinsic tumor cell toxicity. These findings collectively demonstrate that TP5’s antitumor efficacy is entirely dependent on functional T cell immunity, with no direct cytotoxic effects on cancer cells.

### 3.3. TP5 Promotes Thymus Rejuvenation and Counteracts Cancer-Associated Immunosuppression

The thymus serves as the primary site of T cell maturation and selection yet undergoes progressive involution during malignancy and cytotoxic chemotherapy, compromising adaptive immune responses [16]. After treating mice with TP5, we observed that TP5 can promote thymus growth. Notably, the thymus size was increased in the TP5-treated mice compared to the control mice (Figure 3A–C). This thymic enlargement suggested that TP5 actively promotes thymic regeneration in the cancer-bearing state, potentially counteracting tumor-induced thymic atrophy. Notably, we found that the number of T cells, including CD4^+^ T cells, CD8^+^ T cells, and CD4^+^CD8^+^ T cells, were significantly increased in the thymus after the TP5 treatment compared to the control mice (Figure 3D). This is consistent with the findings that thymus rejuvenation is essential for T cell reconstitution in cancer [16]. These results showed that the TP5 treatment in cancer can rejuvenate the thymic atrophy and promote T cell generation in the thymus.

To definitively establish the thymoprotective capacity of TP5, we employed a cyclophosphamide-induced immunosuppression model, which recapitulates the severe thymic involution observed in cancer patients undergoing chemotherapy. The cyclophosphamide administration produced marked thymic atrophy, as evidenced by substantial reductions in organ mass. Remarkably, the concurrent TP5 treatment not only prevented cyclophosphamide-induced thymic regression but actively promoted thymic recovery, resulting in a significantly enhanced thymic mass compared to cyclophosphamide-only controls (Figure 3E–G). Similarly, we also observed that the TP5 treatment significantly increased the population of T cells, including CD4^+^ and CD4^+^CD8^+^ T cells in the thymus, compared to the cyclophosphamide-only control group (Figure 3H). The CD8^+^ T cells also showed an increasing trend compared to the cyclophosphamide-only control but did not reach statistical significance (*p* = 0.18; Figure 3H).

Overall, these findings demonstrate that TP5 possesses potent thymoregenerative properties, capable of both promoting thymic growth in cancer-bearing hosts and protecting against chemotherapy-induced thymic damage. This dual capacity to enhance thymic function while simultaneously combating tumor progression positions TP5 as a unique immunotherapeutic agent that addresses the fundamental immunological imbalance underlying cancer-associated immunosuppression.

### 3.4. TP5 Protects Against T Cell Exhaustion and Restores Effector Function

Having established TP5’s T cell-dependent antitumor mechanisms, we investigated its impact on T cell functional states, particularly its capacity to counteract T cell exhaustion—a fundamental barrier to effective cancer immunotherapy [4]. Within the immunosuppressive tumor microenvironment, T cells progressively upregulate inhibitory receptors including PD-1 and TIM-3, resulting in compromised cytotoxic capacity and diminished effector cytokine production [4,17]. The flow cytometric analysis of splenic T cell populations from tumor-bearing mice revealed that the TP5 treatment drove significant reductions in exhausted T cell phenotypes compared to the vehicle control group, particularly the PD-1^+^TIM-3^+^ double-positive subpopulation (a hallmark of severe, irreversible exhaustion) (Figure 4A–F). This phenotypic restoration suggested that TP5 actively prevents or reverses T cell exhaustion in the cancer setting [4]. To assess functional correlates of this phenotypic improvement, we evaluated the cytokine production capacity—a critical determinant of T cell-mediated tumor control. Ex vivo stimulation assays demonstrated that the TP5 supplementation substantially enhanced the IFN-γ and TNF-α secretion by CD8^+^ T cells (Figure 4G,H), indicating restored effector functionality.

We further validated these findings using the cyclophosphamide-induced immunosuppression model, which allowed for the assessment of TP5’s immunomodulatory effects independent of its thymoregenerative properties. We also found that the TP5 administration in cyclophosphamide-induced immunosuppressive mice significantly augmented T cell effector functions beyond its thymopoietic effects. Intracellular cytokine staining revealed that the TP5 treatment substantially increased the proportion of IFN-γ- and TNF-α-producing CD8^+^ T cells compared to the control animals (Figure 4I,J). Additionally, TP5 enhanced the IL-2 production within thymic T cell populations (Figure 4K). Together, these results showed that TP5 had an essential role in promoting cytokine production and protected against T cell exhaustion.

### 3.5. TP5 Synergistically Enhances the Efficacy of Adoptive T Cell Therapy

To evaluate the clinical translational potential of TP5 as an immunotherapeutic adjuvant in cancer therapy, we assessed its capacity to improve adoptive T cell transfer efficacy using an established B16-OVA melanoma model. Mice were inoculated with B16-OVA cells and subjected to lymphodepleting sub-lethal irradiation before receiving the OT-1 T cell transfer either alone (control) or in combination with the TP5 treatment (Figure 5A). Longitudinal tumor volume measurements revealed striking therapeutic synergy between the TP5 and adoptive T cell transfer. While control animals receiving OT-1 T cells alone showed progressive tumor growth, TP5-treated mice exhibited markedly reduced tumor burdens throughout the observation period (OT-1 T cell vs. OT-1 T cell + TP5: 368.64 ± 130.24 mm^3^ vs. 170.54 ± 30.81 mm^3^, *p* < 0.05; Figure 5B). The therapeutic benefit of this combinatorial approach was further confirmed by a terminal tumor mass analysis, which demonstrated a substantial reduction in the final tumor weight in the TP5-treated group compared to controls (Figure 5C,D). These findings demonstrate that TP5 functions as a potent immunological adjuvant capable of synergistically enhancing adoptive T cell therapeutic outcomes through enhanced immune reconstitution and functional restoration.

## 4. Discussion

Malignant transformation fundamentally disrupts the immunological equilibrium between the host defense and tumor progression, establishing an immunosuppressive microenvironment that systematically compromises the T cell metabolic integrity and effector functionality [18,19]. This hostile tumor milieu imposes severe constraints on T cell immunity through multiple convergent mechanisms, including progressive exhaustion and a diminished cytokine production capacity. These two hallmark deficiencies collectively cripple antitumor surveillance, rendering T cells unable to mount sustained responses against malignant cells [19]. Although contemporary immunotherapeutic strategies, including immune checkpoint blockades, cytokine-based interventions, and combination regimens, have achieved some clinical advances [4], many patients exhibit no response to these therapies, attributed to persistent T cell dysfunction. Therefore, identifying novel immunomodulatory agents capable of restoring T cell competency represents a paramount therapeutic imperative.

Our investigation demonstrates that TP5 exerts potent broad-spectrum antitumor activity through mechanisms inherently dependent on T cell-mediated immunity. It has been reported that TP5 mediates its immunomodulatory effects through interactions with specific cell surface receptors, including the thymopoietin receptor complex and potentially other pattern recognition receptors expressed on T lymphocytes. These interactions enable TP5 to directly modulate intracellular signaling cascades that govern core T cell processes, such as activation, proliferation, and differentiation [20]. Additionally, TP5’s molecular composition, comprising amino acid residues critical for T cell biology [13], also provided a compelling rationale for investigating its immunomodulatory potential. For example, intracellular arginine availability serves as a fundamental determinant of T cell metabolic fitness, with deficiency resulting in functional impairment, while supplementation enhances survival capacity and antitumor efficacy [21,22]. This metabolic dependency, combined with TP5’s established roles in immunoregulation and T cell development [9,10], suggested that TP5 might function as an endogenous immunostimulant capable of reversing cancer-induced T cell dysfunction.

Moreover, recent studies have demonstrated that TP5 promotes the generation of the T cell lineage from human embryonic stem cells [10]. TP5 improves IL-2 production by human lymphocytes [23]. Our study, employing multiple murine cancer models spanning diverse histological origins, revealed a consistent antitumor efficacy that was entirely abrogated in T cell-deficient hosts. The strict immunological dependency of TP5, coupled with the absence of direct cytotoxic effects on malignant cells, definitively established TP5’s mechanism as immunomodulatory rather than cytotoxic. The mechanistic investigation further revealed that the TP5 treatment drives the enhancement of splenic IFN-γ, TNF-α, and IL-2 production, three pivotal immune mediators that collectively shape the magnitude and durability of antitumor immunity, consistent with previous observations of TP5’s capacity to augment IL-2 and IFN-γ secretion [23,24]. Specifically, IFN-γ, as a key driver of the antigen presentation, creates a “priming-friendly” microenvironment that enables naive T cells to recognize tumor antigens [25]. TNF-α, in turn, supports the formation of tertiary lymphoid structures within tumors and enhances endothelial adhesion molecule expression, facilitating the infiltration of activated T cells from the spleen into the tumor microenvironment—overcoming the “infiltration barrier” that often limits the efficacy of T cell-based therapies [26,27,28]. As a central cytokine for T cell expansion and memory formation, IL-2 ensures that antigen-primed T cells (activated via IFN-γ-mediated signals) can proliferate robustly and differentiate into long-lived memory subsets, preventing premature T cell exhaustion and sustaining antitumor responses beyond initial activation [29]. TP5 thus enables improved tumor control, a mechanism independent of T cell function. In addition, TP5 suppressed inhibitory receptor expression, particularly PD-l and TIM-3, critical mediators of T cell exhaustion that progressively compromise the cytotoxic potential, metabolic fitness, and effector molecule production [30,31]. These results demonstrated that TP5 can prevent inhibitory expression and help T cells maintain their effector functions.

Perhaps most significantly, we identified TP5’s unique capacity to promote thymic rejuvenation, a phenomenon with profound implications for cancer immunotherapy. The thymus plays a crucial role in T lymphocyte development by offering a specialized microenvironment that supports the maturation of T cell lymphocytes [32,33] yet undergoes progressive involution with aging and accelerated atrophy during malignancy, resulting in a compromised T cell output and inadequate tumor surveillance [34,35]. Interestingly, our demonstration that TP5 not only prevents cancer-associated thymic regression but actively promotes regeneration in chemotherapy-induced immunosuppression models suggests a fundamental mechanism addressing the immunological imbalance underlying cancer progression.

These mechanistic insights hold profound therapeutic significance for clinical cancer immunotherapy. Unlike some immunomodulatory agents that target a single bottleneck in antitumor immunity, TP5 addresses multiple, interconnected barriers that currently limit immunotherapeutic efficacy: it enhances T cell effector functions (e.g., cytotoxicity and cytokine secretion), mitigates exhaustion-associated dysfunction (a primary driver of therapy resistance), and promotes thymic regeneration (critical for sustaining long-term T cell repertoire diversity). This multifaceted mode of action positions TP5 as a uniquely versatile candidate for advancing immunotherapy. Notably, our demonstration of TP5’s synergistic activity with adoptive T cell transfer particularly supports TP5’s potential as a clinical adjuvant, capable of amplifying therapeutic responses while potentially overcoming resistance mechanisms that constrain current immunotherapies. Future investigations should elucidate the molecular pathways governing TP5-mediated thymic rejuvenation and assess its translational potential in clinical cancer immunotherapy.

A major limitation of TP5 for clinical application is its extremely short plasma half-life of approximately 30 s in humans, which is primarily due to the rapid enzymatic degradation by serum peptidases. This pharmacokinetic profile raises important questions about tissue penetration and the site of action for TP5’s immunomodulatory effects. However, biodistribution studies using FITC-conjugated TP5 have provided valuable insights, demonstrating that despite its brief circulation time, TP5 can reach various tissues, including lymphoid organs and tumor sites, and, remarkably, can cross the blood–brain barrier to accumulate in brain tissue, with therapeutic efficacy validated in preclinical glioblastoma models [36]. Nevertheless, extending TP5’s half-life represents a critical priority for optimizing its clinical utility, and several formulation strategies have been developed or proposed to address this limitation, including nanoparticle encapsulation (liposomal and polymeric carriers), to protect TP5 from enzymatic degradation while enabling controlled release and potentially enhancing tumor accumulation [37]; chemical modification strategies, such as using the albumin binding strategy to reduce renal clearance [38]; and sustained-release depot formulations to provide continuous low-level exposure for maintaining chronic immune activation [39]. Future studies could explore more promising strategies that balance half-life extension with the preservation of biological activity and safety, as the development of long-acting TP5 formulations could significantly enhance its clinical applicability and therapeutic efficacy in cancer immunotherapy.

While the present study demonstrates that TP5 exerts antitumor effects by suppressing tumor growth, protecting T cells from exhaustion, and promoting thymic rejuvenation with an additional enhancement of adoptive T cell therapy, several limitations should be acknowledged to guide future research. Our current mechanistic insights into TP5’s actions remain focused on its role in thymic rejuvenation, such as increasing thymic T cell numbers and supporting thymic morphological recovery; however, the molecular and cellular pathways underlying TP5-mediated thymic development (e.g., whether TP5 regulates key transcription factors, signaling cascades, or stromal cell function in the thymus) have not yet been fully elucidated. Additionally, while thymic rejuvenation is linked to improved antitumor immunity in our models, the specific mechanisms by which restored thymic function modulates thymus–cancer crosstalk (e.g., how thymus-derived T cells traffic to the tumor microenvironment, or how thymic signals interact with tumor-infiltrating immune cells) remain unclear and require further investigation. These limitations do not diminish the core findings of TP5’s therapeutic potential in cancer models but highlight critical directions for future work to advance our understanding of TP5’s role in antitumor immunity.

Collectively, our findings establish TP5 as a multifaceted immunomodulator that enhances T cell-mediated antitumor immunity by promoting effector functionality, preventing exhaustion-associated dysfunction, and facilitating thymic rejuvenation (Figure 6). Notably, adoptive transfer studies demonstrated that TP5-conditioned T cells achieved markedly superior antitumor efficacy compared to conventional preparations, establishing TP5’s potential as a clinical adjuvant. Thus, TP5 represents a promising therapeutic strategy that transcends current single-target interventions, offering transformative potential for cancer immunotherapy across diverse malignancies.

## 5. Conclusions

TP5 is an immunostimulant drug that has been widely used in the treatment of immunoregulatory diseases. However, its role in modulating T cells in cancer has not been previously studied. This study found that TP5 effectively controls the growth of multiple tumors by promoting cytokine production, preventing T cell exhaustion, and enhancing adoptive T cell therapy. Notably, the antitumor effect of TP5 entirely depends on T cells. Altogether, these findings demonstrate that TP5 could be a potent adjuvant in T cell-based cancer therapy, either alone or in combination with other treatment approaches.

## Figures and Tables

**Figure 2 biomedicines-13-02494-f002:**
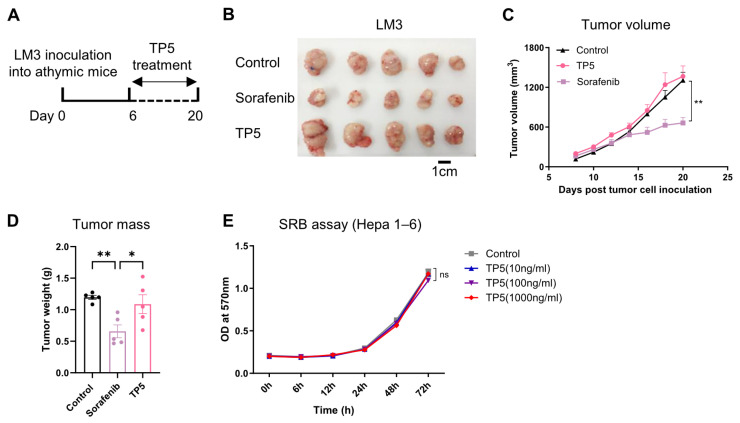
The antitumor activity of TP5 relies on T cells. (**A**–**D**) BALB/c nu/nu athymic nude mice were inoculated with LM3 hepatocellular carcinoma cells (1 × 10^6^ cells/mouse) for the subcutaneous hepatoma model and treated with TP5 (20 mg/kg/day) or sorafenib (30 mg/kg/day): (**A**) experimental timeline, (**B**) representative tumors, (**C**) tumor progression of LM3, and (**D**) tumor mass on day 20 (*n* = 5 mice per group for (**A**–**D**). (**E**) cell viability assays were performed by SRB. Hepa 1-6 tumor cells were treated with different concentrations of TP5 (10 ng/mL, 100 ng/mL, and 1000 ng/mL), and cell viability was measured at 0 h, 6 h, 12 h, 24 h, 48 h, and 72 h (*n* = 5 per group). *p* values were calculated using ordinary two-way analysis of variance (ANOVA) for (**C**,**E**) and one-way (ANOVA) for (bar graphs in (**D**)). Data presented as mean ± SEM. * *p* < 0.05 and ** *p* < 0.01. ns means not significant.

**Figure 3 biomedicines-13-02494-f003:**
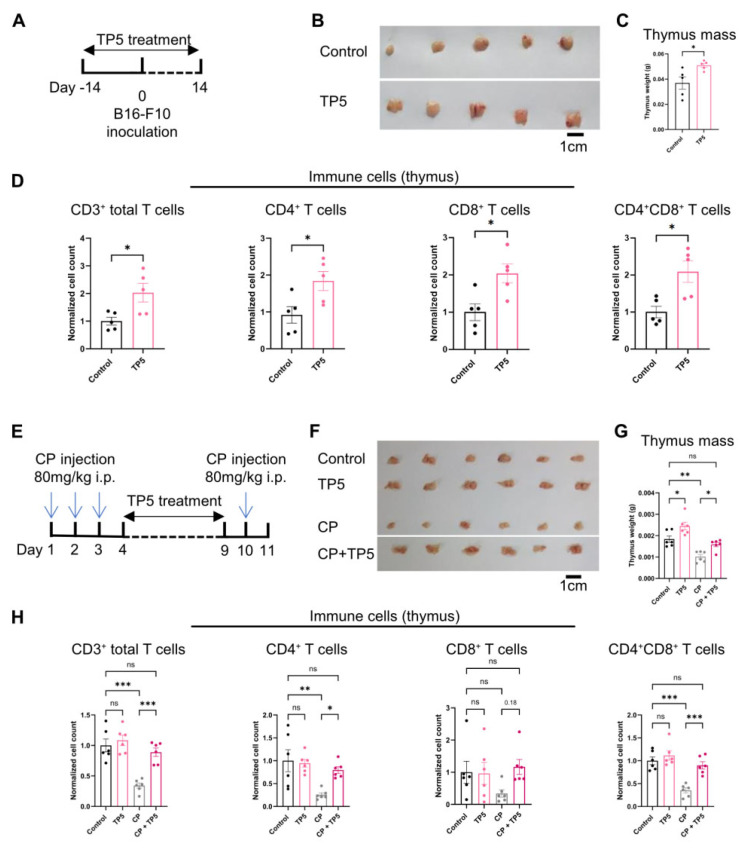
TP5 promotes thymus development in cancer. (**A**–**D**) after treating tumor-bearing mice with TP5 (20 mg/kg/day) for 14 days, the thymus was collected and weighted: (**A**) experimental timeline, (**B**) representative thymus, (**C**) thymus mass on day 14, and (**D**) T cell population in thymus on day 14 (cell numbers were normalized to the average of control) (*n* = 5 mice per group for (**A**–**D**)). (**E**–**H**) BALB/c mice were administered cyclophosphamide 80 mg/kg/day via intraperitoneal injection (i.p.) on days 1–3 and day 10. From days 4 to 10, mice were subcutaneously administered TP5 (20 mg/kg/day) or saline, and on day 11, the thymus was collected for analysis: (**D**) experimental timeline, (**E**) representative thymus, (**F**) thymus mass on day 11, and (**H**) T cell population in thymus on day 11 (cell numbers were normalized to the average of control) (*n* = 6 mice per group for (**E**–**H**)). *p* values were calculated using unpaired two-tailed *t*-tests for (bar graphs in (**C**,**D**)) and one-way (ANOVA) for (bar graphs in (**F**,**H**)). Data presented as mean ± SEM. * *p* < 0.05 and ** *p* < 0.01 and *** *p* < 0.001. ns means not significant.

**Figure 4 biomedicines-13-02494-f004:**
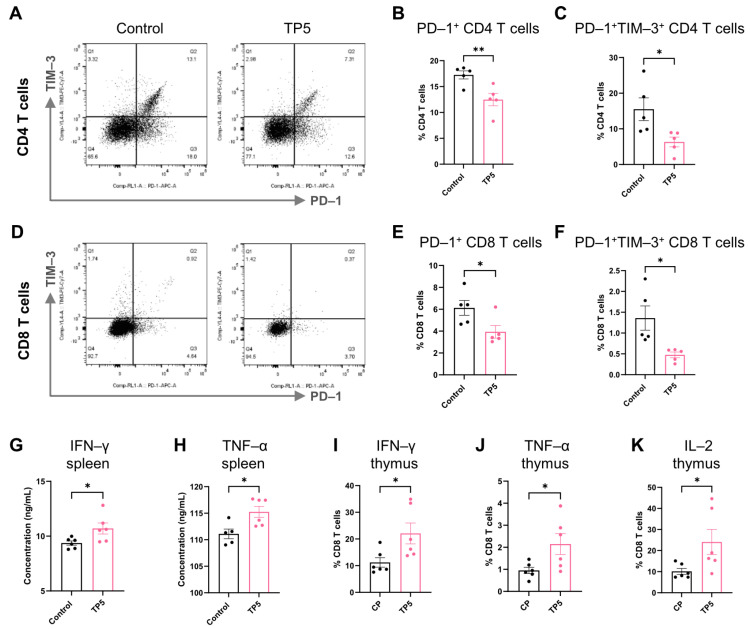
TP5 protects against T cell exhaustion and promotes cytokine production. (**A**–**F**) splenocytes were isolated from B16-F10 tumor-bearing mice after treating with TP5 on 14 days post-tumor inoculation, and T cell inhibitory receptors PD-1 and TIM-3 expressed on CD4^+^ and CD8^+^ T cells were analyzed by flow cytometry: (**A**,**D**) representative FACS analysis of the frequency of PD-1- and TIM-3-positive CD4^+^ T cells and CD8^+^ T cells, respectively; (**B**) frequency of PD-1 expression by CD4^+^ T cells; (**C**) frequency of PD-1 and TIM-3 expression by CD4^+^ T cells; (**E**) frequency of PD-1 expression by CD8^+^ T cells; and (**F**) frequency of PD-1 and TIM-3 expression by CD8^+^ T cells in the spleens (*n* = 5 mice per group for (**A**–**F**)). (**G**,**H**) the CD8^+^ T cell culture media were supplemented with TP5 (1 μg/mL), and IFN-γ and TNF-α production was determined by ELISA of the CD8^+^ T cell culture supernatant on day 5 (*n* = 6 mice per group). (**I**–**K**) CD8^+^ single positive T cells were isolated from TP5-treated cyclophosphamide-induced immunosuppressive mice and stained with cytokine markers (IFN-γ, TNF-α, and IL-2) (*n* = 6 mice per group). *p* values were calculated using unpaired two-tailed *t*-tests for (bar graphs in (**B**,**C**,**E**,**F**–**K**)). Data presented as mean ± SEM. * *p* < 0.05 and ** *p* < 0.01.

**Figure 5 biomedicines-13-02494-f005:**
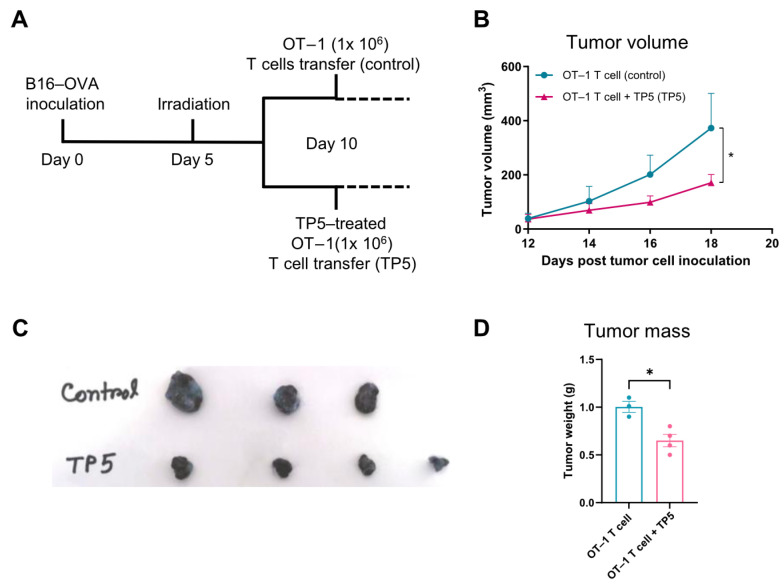
TP5 promotes the antitumor effect of T cell therapy. (**A**–**D**) OT–1 T cells were expanded with TP5 (1 μg/mL) before transferring to tumor-bearing mice, on day 5 mice were irradiated, and on day 10 TP5–treated or untreated OT–1 T cells (1 × 10^6^ per mouse) were transferred to B16–OVA tumor-bearing mice: (**A**) experimental timeline, (**B**) B16–OVA melanoma tumor progression, (**C**) representative tumors, and (**D**) tumor mass on day 18 (*n* = 3–4 mice per group for (**A**–**D**)). *p* values were calculated using ordinary two-way analysis of variance (ANOVA) for (**B**) and unpaired two-tailed *t*-tests for (bar graphs in (**D**)). Data presented as mean ± SEM. * *p* < 0.05.

**Figure 6 biomedicines-13-02494-f006:**
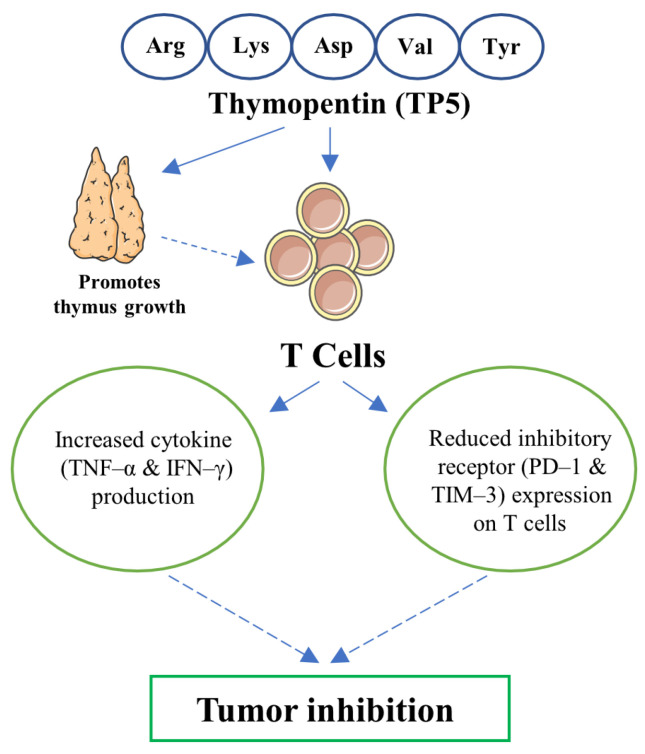
Thymopentin (TP5) promotes anti-tumor immunity through thymic rejuvenation and T cell functional enhancement. Thymopentin, a pentapeptide comprising Arg-Lys-Asp-Val-Tyr, exerts dual mechanisms in tumor immunotherapy. TP5 promotes thymus growth and regeneration, supporting de novo T cell development and maintaining a diverse T cell repertoire. Simultaneously, TP5 enhances T cell effector functions by augmenting the production of key pro-inflammatory cytokines, including tumor necrosis factor-α (TNF-α) and interferon-γ (IFN-γ), which are critical mediators of anti-tumor immunity. Additionally, TP5 prevents T cell exhaustion by reducing the expression of inhibitory checkpoint receptors programmed cell death protein 1 (PD-1) and T cell immunoglobulin and mucin-domain containing-3 (TIM-3) on T cells. Through these coordinated mechanisms (thymic rejuvenation, preservation of effector function, and prevention of exhaustion), TP5 sustains functional T cell responses that collectively contribute to effective tumor growth inhibition. Arrows indicate promoting effects; dashed arrows indicate downstream consequences.

## Data Availability

The data will be available upon request to the corresponding authors.

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
