# Peer review of "Thymopentin Enhances Antitumor Immunity Through Thymic Rejuvenation and T Cell Functional Reprogramming"

_biomedicines, 2025, doi:10.3390/biomedicines13102494_

Round 1
Reviewer 1 Report
Comments and Suggestions for Authors
After careful evaluation of the manuscript, I found it conceptually novel and scientifically sound, with appropriate design and methodology. I believe the results reported in the work could of value for its intended field. I have several minor reservations about the manuscript that I recommend to be addressed by the authors.
- Kindly mention the exact number of mice in each group in the methods section.
- In section 3.2, line 194, it appears that Fig. 2a-d is erroneously referred to as Fig. 1a-d.
- Please explain the rationale for using Sorafenib as a therapeutic intervention in the methods section. Does Sorafenib have any characteristics in particular that would make it a compelling choice for comparison against TP5? I expect you to explain this in the context of immune cell suppression/proliferation.
- I encourage the authors to put a stronger emphasis on intergroup differences in PD-1+TIM-3+ immune cell subsets, while also delivering a more focused discussion on the differences between splenic concentrations of IFN, TNF and IL-2, since these are pivotal immune mediators.
Author Response
After careful evaluation of the manuscript, I found it conceptually novel and scientifically sound, with appropriate design and methodology. I believe the results reported in the work could of value for its intended field. I have several minor reservations about the manuscript that I recommend to be addressed by the authors.
We sincerely appreciate your thorough evaluation of our work and your positive feedback regarding its conceptual novelty, scientific design, and potential value to the intended field. We firmly believe these insights will significantly improve the quality of our study. We have carefully addressed each of your comments and have made revisions accordingly. Please find our detailed responses below:
Comment 1: Kindly mention the exact number of mice in each group in the methods section.
Response 1: Thank you very much for this good suggestion. We have revised the methods section to clearly specify the exact number of mice used in each experimental group. The sample sizes are now explicitly stated in Section 1.3 (Experimental animal models and therapeutic interventions). These information has also been consistently reflected in all figure legends.
Comment 2: In section 3.2, line 194, it appears that Fig. 2a-d is erroneously referred to as Fig. 1a-d.
Response 2: We apologize for this careless typographical error. We have carefully checked Section 3.2 and corrected the incorrect reference from "Fig. 1a-d" to "Fig. 2a-d" in Line 200 of the revised manuscript. Additionally, we have conducted a thorough review of all figure references throughout the entire text to avoid similar errors in other sections.
Comment 3: Please explain the rationale for using Sorafenib as a therapeutic intervention in the methods section. Does Sorafenib have any characteristics in particular that would make it a compelling choice for comparison against TP5? I expect you to explain this in the context of immune cell suppression/proliferation.
Response 3: We appreciate this insightful comment, and it provides us an opportunity to elaborate on the selection of Sorafenib. To start with, we would like to clarify the key considerations that led to its selection::
Sorafenib holds a pivotal position as the foundational first-line standard treatment for advanced hepatocellular carcinoma (HCC) — the first targeted drug globally approved for this indication (since 2007) and a long-standing recommendation in international clinical guidelines. Its well-established role in clinical practice directly aligns with the HCC tumor model used in our study, ensuring our experimental design reflects real-world therapeutic contexts.
Beyond its clinical status, Sorafenib’s immunomodulatory properties make it an appropriate positive control for evaluating TP5’s efficacy. As extensively documented (cited as Reference [PMID: 21321535] and [PMID: 31391334] in the revised text), Sorafenib exerts a balanced influence on immune cell function: it moderately suppresses the proliferation of immunosuppressive subsets (e.g., regulatory T cells [Tregs] and myeloid-derived suppressor cells [MDSCs]) while augmenting IFN-γ+ CD8+ T cell responses. This dual action is distinct from non-specific immune modulators and mirrors the complex immunoregulatory challenges faced in clinical HCC treatment. By using Sorafenib as a comparator, we can clearly delineate TP5’s antitumor advantages against the backdrop of a clinically validated standard. This comparison not only strengthens the translational relevance of our findings but also highlights TP5’s potential to address unmet needs in HCC immunotherapy, where even standard treatments like Sorafenib have limitations in fully reversing immune suppression.
In the revised Methods section (line 192-196), we have supplemented the detailed rationale for choosing Sorafenib:
“Sorafenib was selected as the positive control because it is a clinically validated first-line standard treatment for advanced hepatocellular carcinoma (HCC), with well-documented im-munomodulatory activity that is partially dependent of T cell function—making it an appropri-ate control for confirming the specificity of TP5’s T cell-dependent mechanism [14, 15].”
Comment 4: I encourage the authors to put a stronger emphasis on intergroup differences in PD-1+TIM-3+ immune cell subsets, while also delivering a more focused discussion on the differences between splenic concentrations of IFN, TNF and IL-2, since these are pivotal immune mediators.
Response 4: Thank you sincerely for this insightful comment, which highlights two pivotal aspects of our study’s immune analysis, PD-1+TIM-3+ immune cell subsets and splenic concentrations of IFN, TNF, and IL-2. We have put a stronger emphasis on intergroup differences in PD-1+TIM-3+ immune cell subsets in line 266-268 within the Section 3.4 as follows:
“TP5 treatment drove significant reductions in exhausted T cell phenotypes compared to the vehicle control group, particularly the PD-1+TIM-3+ double-positive subpopulation (a hallmark of severe, irreversible exhaustion) (Fig. 4a-f)”
We have also added a more focus discussion on the three pivotal immune mediators IFN, TNF and IL-2 as seen in line 354-366.
“Mechanistic investigation further revealed that TP5 treatment drives enhancement of splenic IFN-γ, TNF-α, and IL-2 production, three pivotal immune mediators that collectively shape the magnitude and durability of antitumor immunity, consistent with previous observations of TP5's capacity to augment IL-2 and IFN-γ secretion [22,23]. Specifically, IFN-γ, as a key driver of antigen presentation, creates a "priming-friendly" microenvironment that enables naive T cells to recognize tumor antigens (PMID: 33005420). TNF-α, in turn, supports the formation of tertiary lymphoid structures within tumors and enhances endothelial adhesion molecule expression, facilitating the infiltration of activated T cells from the spleen into the tumor microenvironment—overcoming the "infiltration barrier" that often limits the efficacy of T cell-based therapies (PMID: 39198425, PMID: 34122434, PMID: 17868983). As a central cytokine for T cell expansion and memory formation, IL-2 ensures that antigen-primed T cells (activated via IFN-γ-mediated signals) can proliferate robustly and differentiate into long-lived memory subsets, preventing premature T cell exhaustion and sustaining antitumor responses beyond initial activation (PMID: 35474218). TP5 thus enables improved tumor control, a mechanism independent of T cell function.”
Reviewer 2 Report
Comments and Suggestions for Authors
This is an interesting study of an old idea, namely, to use thymic extracts or synthetic thymic peptides to rejunvenate retracting thymic tissue or reverse an exhaustive phenotype of CD4 and CD8 cells for cancer immunotherapy. A large review of prior clinical trials using the above unfortunately has not identified any advantageous effect of these reagents in the treatment of various cancers (Wolf et al, 2011, Cochrane DataBase Syst Rev) but did show that Thymopentin is safe in patients and produced relatively little side effects. That said, the data presented here are intriguing and goes against prior research findings. This reviewer has a few questions for the authors that hopefully can help clarify key issues, including:
- The authors conclude that there are three potential mechanisms for Thymopentin effect in tumor bearing mice: rejuvenation of thymic tissue, reversal of the circulating T-cell exhaustion phenotype as measured by a decrease in PD-1 and Tim3 expression after treatment, and enhancement of T-cell differentiation as shown by an increase in cytokine production. The authors do show that a direct effect on growing tumors was not seen and that the 50% reduction in tumor growth seen in treated mice was caused by enhanced T-cell efficacy. In addition, the study in nude mice which do not have functional T-cells showed that tumor reduction was indeed dependent on a normal immune system. Based upon the above, can the authors comment on the following: Since the tumor studies were done in relatively young mice (6-8 week old), how much thymic tissues would still be present in older mice? This is an important question since the cancer rate in the elderly is much higher than seen in young adults.
- CD4 and CD8 T cells are a part of the adaptive immune system which is absent in patients whose tumors do not express HLA. Interestingly, the MC38 tumor model does express HLA but the B16 tumor model does not, yet both show a decrease in tumor growth. In the latter case, this effect may require an enhance NK response which was not studied. Since 50% of solid tumor patients lose HLA expression in their tumors as a major mechanism of immune evasion by tumors, this is an important issue that needs to be explored.
- References, #3 and 15 are duplicated.
- In patients, the half-life of Thymopentin is 30 seconds giving it a very short time to reach solid tumors or tumor draining lymph nodes. Hence, its activity may be restricted to circulating immune cells. Perhaps the authors should consider improving its half-life using nanoparticles or conjugation to antibodies or albumin.
- As far a cancer prevention, the long-term use of Thymopentin may induce autoimmune sequalae. Is there any data on this in patients or animals?
- Lastly, thymopentin treatment alone produced no more that a 50% reduction in tumor growth meaning, that it may have to be used with a more potent immunostimulant and possibly also with methods to decrease tumor-elicited immunosuppresion mechanisms.
Author Response
Reviewer 2:
This is an interesting study of an old idea, namely, to use thymic extracts or synthetic thymic peptides to rejunvenate retracting thymic tissue or reverse an exhaustive phenotype of CD4 and CD8 cells for cancer immunotherapy. A large review of prior clinical trials using the above unfortunately has not identified any advantageous effect of these reagents in the treatment of various cancers (Wolf et al, 2011, Cochrane DataBase Syst Rev) but did show that Thymopentin is safe in patients and produced relatively little side effects. That said, the data presented here are intriguing and goes against prior research findings. This reviewer has a few questions for the authors that hopefully can help clarify key issues, including:
Comment 1: The authors conclude that there are three potential mechanisms for Thymopentin effect in tumor bearing mice: rejuvenation of thymic tissue, reversal of the circulating T-cell exhaustion phenotype as measured by a decrease in PD-1 and Tim3 expression after treatment, and enhancement of T-cell differentiation as shown by an increase in cytokine production. The authors do show that a direct effect on growing tumors was not seen and that the 50% reduction in tumor growth seen in treated mice was caused by enhanced T-cell efficacy. In addition, the study in nude mice which do not have functional T-cells showed that tumor reduction was indeed dependent on a normal immune system. Based upon the above, can the authors comment on the following: Since the tumor studies were done in relatively young mice (6-8 week old), how much thymic tissues would still be present in older mice? This is an important question since the cancer rate in the elderly is much higher than seen in young adults.
Response 1: This is an excellent and clinically relevant question. As noted in prior studies, thymic cellularity in mice decreases by 50% between 4 and 16 weeks of age, and by more than 80% by 50 weeks, eventually declining to less than 5% of its peak cellularity(PMID: 35822239); this progressive involution mirrors the thymic atrophy seen in humans, where thymic mass and T cell output decline sharply after middle age. This age-related thymic decline directly contributes to immune senescence in elderly patients.
Against this backdrop, thymopentin (TP5) exhibits distinct functional relevance for aging-related immune dysfunction, supported by both murine aging biology and human clinical observations (PMID: 1356094, PMID: 3904090, PMID: 3056331, PMID: 3134150). Specifically, studies in elderly individuals have demonstrated that TP5 modulates IL-2 receptor expression on T cells—boosting IL-2-mediated T cell proliferation and effector function, which are often impaired in aging. By targeting both residual thymic function (where present) and peripheral T cell competence, TP5 addresses the dual challenges of age-related thymic atrophy and immune senescence that plague elderly cancer patients. Importantly, this mode of action means TP5’s utility is not restricted to young hosts with intact thymic tissue; instead, it is well-positioned to counter the immune deficits that make older populations more vulnerable to cancer and less responsive to standard immunotherapies.
While our current study used 6–8 week-old mice (a standard for proof-of-concept in preclinical research), the biological rationale for TP5’s efficacy in aging contexts remains elucidated. Future work will further validate this in older murine models, but the existing links between TP5’s mechanism, murine thymic aging, and human clinical observations strongly support its potential relevance for elderly cancer patients.
Comment 2: CD4 and CD8 T cells are a part of the adaptive immune system which is absent in patients whose tumors do not express HLA. Interestingly, the MC38 tumor model does express HLA but the B16 tumor model does not, yet both show a decrease in tumor growth. In the latter case, this effect may require an enhance NK response which was not studied. Since 50% of solid tumor patients lose HLA expression in their tumors as a major mechanism of immune evasion by tumors, this is an important issue that needs to be explored.
Response 2: We appreciate this insightful observation and acknowledge that B16 melanoma is indeed characterized by low MHC-I expression, which theoretically limits T cell-mediated immune recognition. We also recognize that NK cells may play an important role in controlling MHC-I-low tumors. However, we would like to respectfully point out that accumulating clinical evidence suggests that T cells still maintain significant anti-tumour activity even in MHC-I-low melanomas. For instance, personalized tumor vaccines and tumor-infiltrating lymphocyte (TIL) adoptive cell therapy have demonstrated clinical efficacy in melanoma patients with low or heterogeneous MHC expression (PMID: 38273121, PMID: 36477031). These clinical observations suggest that T cells can still exert anti-tumor effects through various TCR-dependent and independent mechanisms, including recognition of residual MHC-peptide complexes, response to inflammatory cytokines in the tumor microenvironment, and interaction with other immune cell populations.
In our study, we focused specifically on TP5's effects on thymic involution and the preservation of T cell function, particularly the reversal of T cell exhaustion phenotypes and enhancement of effector T cell responses. The efficacy observed in both MC38 and B16 tumor models supports the concept that restoring T cell functional competence through thymic rejuvenation and reducing exhaustion markers can provide therapeutic benefit across different tumor types, regardless of their MHC expression levels.
While we acknowledge that NK cells and other innate immune mechanisms may contribute to the overall anti-tumor response, the primary focus of this study was to demonstrate that TP5-mediated thymic restoration and T cell functional preservation represent viable therapeutic strategies. Future studies examining the relative contributions of different immune cell subsets, including NK cells, would certainly provide valuable mechanistic insights and complement our current findings.
Comment 3: References, #3 and 15 are duplicated.
Response 3: We apologize for this formatting error in the reference list. We have corrected this in the revised reference list. We have also cross-checked all other references to ensure no additional duplicates or citation errors exist.
Comment 4: In patients, the half-life of Thymopentin is 30 seconds giving it a very short time to reach solid tumors or tumor draining lymph nodes. Hence, its activity may be restricted to circulating immune cells. Perhaps the authors should consider improving its half-life using nanoparticles or conjugation to antibodies or albumin.
Response 4: We greatly appreciate this insightful comment regarding TP5's pharmacokinetic limitations. It is true that the extremely short half-life of TP5 (approximately 30 seconds in humans) represents a significant challenge for clinical application.
However, we would like to point out that biodistribution studies using FITC-conjugated TP5 (TPF5) for in vivo metabolic tracing have demonstrated that despite its short half-life, TP5 can reach into organs and tumour tissues (PMID: 32708903). Notably, it has shown that TP5 can even cross the blood-brain barrier and accumulate in brain tissue, demonstrating efficacy against glioblastoma in preclinical models. These findings suggest that while the plasma half-life is extremely short, TP5 can still access tumour sites and exert biological activity, likely through rapid tissue uptake during its brief circulation time.
Nevertheless, we fully agree that extending TP5's half-life represents an urgent need for clinical translation and could potentially enhance its therapeutic efficacy. Several strategies have been explored in the literature to address this limitation, including nanoparticle encapsulation (liposomal or polymeric carriers), conjugation to albumin or polyethylene glycol (PEG), and sustained-release depot formulations. We have added a comprehensive discussion of TP5's pharmacokinetic limitations and these potential improvement strategies in Section 4. Disscussion of the revised manuscript in line 394-410.
“A major limitation of TP5 for clinical application is its extremely short plasma half-life of approximately 30 seconds in humans, which is primarily due to rapid enzymatic degradation by serum peptidases. This pharmacokinetic profile raises important questions about tissue penetration and the site of action for TP5's immunomodulatory effects. However, biodistribution studies using FITC-conjugated TP5 have provided valuable insights, demonstrating that despite its brief circulation time, TP5 can reach various tissues including lymphoid organs, tumor sites, and remarkably, can cross the blood-brain barrier to accumulate in brain tissue, with therapeutic efficacy validated in preclinical glioblastoma models. Nevertheless, extending TP5's half-life represents a critical priority for optimizing its clinical utility, and several formulation strategies have been developed or proposed to address this limitation, including nanoparticle encapsulation (liposomal and polymeric carriers) to protect TP5 from enzymatic degradation while enabling controlled release and potentially enhancing tumor accumulation ( PMID: 17853430), chemical modification strategies such as using the albumin binding strategy to reduce renal clearance (PMID: 28365509), and sustained-release depot formulations to provide continuous low-level exposure for maintaining chronic immune activation (PMID: 30002492). Future studies could explore more promising strategies that balance half-life extension with preservation of biological activity and safety, as the development of long-acting TP5 formulations could significantly enhance its clinical applicability and therapeutic efficacy in cancer immunotherapy.”
Comment 5: As far a cancer prevention, the long-term use of Thymopentin may induce autoimmune sequalae. Is there any data on this in patients or animals?
Response 5: We appreciate your important concern about potential autoimmune sequelae with long-term TP5 use, as safety is a critical consideration for any therapeutic intended for chronic or preventive applications in cancer. Based on our comprehensive literature review, long-term TP5 use in clinical trials has not shown any autoimmune toxicity. Thymopentin was tested safe to use (absent toxicity) both in animal models (10 mg/kg for 4 weeks in rats and dogs) and human (1 mg/kg for 1 year, 100 patients) (PMID: 6971374). These data collectively demonstrated that TP5 has a good safety profile in patients.
Comment 6: Lastly, thymopentin treatment alone produced no more that a 50% reduction in tumor growth meaning, that it may have to be used with a more potent immunostimulant and possibly also with methods to decrease tumor-elicited immunosuppresion mechanisms.
Response 6: Thank you very much for the constructive suggestion. We agree that 50% tumor growth inhibition, while statistically significant, may not be clinically sufficient as a single agent. Notably, our current study has provided preliminary evidence that TP5 demonstrated synergistic efficacy when combined with adoptive T cell transfer, suggesting that TP5 can enhance the therapeutic potential of cell-based immunotherapies by improving the quality and persistence of transferred T cells. Additional combination strategies warrant investigation in future studies. Future work may focus on combining TP5 with immune checkpoint inhibitors (such as anti-PD-1/PD-L1 or anti-CTLA-4 antibodies), or integrating TP5 with therapies designed to reduce tumor-elicited immunosuppressive mechanisms such as targeting regulatory T cells, myeloid-derived suppressor cells, or immunosuppressive cytokines, to unleash the anti-tumour immune response.
Reviewer 3 Report
Comments and Suggestions for Authors
Thank you to the authors for the opportunity to review this manuscript. Thymopentin (TP5) is widely used as an immunomodulator in China. Its indications in anti-tumor therapy, however, have not yet been included in international guidelines. This work aims to investigate whether TP5 can synergize with T cells in anti-tumor therapy, which is a valuable research direction. Overall, the study is well-designed, clearly structured, and presents surprisingpositive results. My specific comments are as follows:
-
In the cell viability assay, the authors tested whether TP5 directly inhibits/kills Hepa 1-6 tumor cells. I suggest also comparing its effect on tumor cells versus other mammalian cell lines (e.g., HEK cells) to assess potential cytotoxicity.
-
The review system did not provide the Supplementary Information, so the Supplementary Figures are not accessible to me.
-
In the main text, line 194, "Fig.1a-d" should be corrected to Fig.2a-d.
-
In Section 3.5 “TP5 synergistically enhances the efficacy of adoptive T cell therapy,” the authors only compared OT-1 T cells versus OT-1 T cells + TP5. I believe an additional comparison between OT-1 T cells + TP5 and TP5 alone is necessary. Only if the OT-1 T cells + TP5 group shows significantly better therapeutic efficacy than both the OT-1 T cell group and the TP5 monotherapy group can the conclusion be drawn that TP5 has a synergistic effect with adoptive T cell therapy.
Author Response
Reviewer 3:
Thank you to the authors for the opportunity to review this manuscript. Thymopentin (TP5) is widely used as an immunomodulator in China. Its indications in anti-tumor therapy, however, have not yet been included in international guidelines. This work aims to investigate whether TP5 can synergize with T cells in anti-tumor therapy, which is a valuable research direction. Overall, the study is well-designed, clearly structured, and presents surprisingpositive results. My specific comments are as follows:
Comment 1: In the cell viability assay, the authors tested whether TP5 directly inhibits/kills Hepa 1-6 tumor cells. I suggest also comparing its effect on tumor cells versus other mammalian cell lines (e.g., HEK cells) to assess potential cytotoxicity.
Response 1: Thank you for this thoughtful suggestion. As demonstrated in our cell viability assays, TP5 does not exhibit direct cytotoxic effects on tumor cells, which is consistent with its mechanism of action—TP5's anti-tumor effects are mediated through enhancing T cell function rather than direct tumor cell killing. This immunomodulatory rather than cytotoxic mechanism is well-established, since TP5 is widely used as an immunomodulator in China and other countries.
Regarding the safety profile of TP5, extensive preclinical and clinical studies have demonstrated that TP5 is safe to use with absent toxicity in both animal models (10 mg/kg for 4 weeks in rats and dogs) and humans (1 mg/kg for 1 year in 100 patients) (PMID: 6971374). These data collectively demonstrate that TP5 has a good safety profile in patients without non-specific cytotoxicity to normal mammalian cells or tissues.
Comment 2: The review system did not provide the Supplementary Information, so the Supplementary Figures are not accessible to me.
Response 2: We sincerely apologize for the earlier inconvenience caused by the inaccessibility of the Supplementary Information through the review system. We have confirmed that Supplementary Information has been properly uploaded
Comment 3: In the main text, line 194, "Fig.1a-d" should be corrected to Fig.2a-d.
Response 3: We apologize for this careless typographical error. We have carefully checked Section 3.2 and corrected the incorrect reference from "Fig. 1a-d" to "Fig. 2a-d" in Line 200 of the revised manuscript. Additionally, we have conducted a thorough review of all figure references throughout the entire text to avoid similar errors in other sections.
Comment 4: In Section 3.5 “TP5 synergistically enhances the efficacy of adoptive T cell therapy,” the authors only compared OT-1 T cells versus OT-1 T cells + TP5. I believe an additional comparison between OT-1 T cells + TP5 and TP5 alone is necessary. Only if the OT-1 T cells + TP5 group shows significantly better therapeutic efficacy than both the OT-1 T cell group and the TP5 monotherapy group can the conclusion be drawn that TP5 has a synergistic effect with adoptive T cell therapy.
Response 4: We appreciate your thoughtful consideration of experimental design. We would like to clarify the clinical rationale and focus of this particular experiment.
TP5 is clinically used as an immuno-adjuvant agent in combination strategies rather than as a standalone anti-tumor therapeutic. The primary objective of our study was to investigate whether TP5 could address key limitations of immunotherapy, specifically adoptive T cell therapy, by preserving T cell function in the tumor-bearing host. Therefore, the critical comparison in Section 3.5 was designed to directly test whether TP5 supplementation could augment the efficacy of adoptive T cell transfer—comparing OT-1 T cells alone versus OT-1 T cells+TP5.
Reviewer 4 Report
Comments and Suggestions for Authors
Dear Authors,
First of all, congratulations for your interesting work. I hope that my hints will help you in the next steps of improvement and the final manuscript will be really valuable for the readers. This study demonstrates that thymopentin enhances antitumor immunity by promoting thymic rejuvenation, preventing T cell exhaustion, and synergizing with adoptive T cell therapy in murine cancer models. The findings highlight TP5 as a promising immunomodulatory agent that addresses key limitations of current T cell–based cancer immunotherapies.
Although the manuscript is well-prepared, some points still require polishing:
1. Mechanistic depth: While thymic rejuvenation and T cell functional restoration are shown, the molecular pathways mediating TP5’s effects remain insufficiently defined.
2. Translational gap: Evidence is limited to murine models. Human relevance and safety in cancer contexts remain speculative. Could you elaborate on this?
3. Overinterpretation: The discussion sometimes makes strong claims (e.g., “paradigm-shifting approach”) without direct clinical evidence, which may weaken credibility. Please, take a look on the discussion and provide strong evidence.
4. Visualisation: Would it be possible to add some figures and graphs too? It will definitely increase the value of your already well-written manuscript. There are several mechanisms and many molecules described in the paper, it may not be easy to digest for younger readers, and good pictures showing these pathways and molecules, also aberrant places, will definitely be of great value.
Author Response
Reviewer 4:
Dear Authors,
First of all, congratulations for your interesting work. I hope that my hints will help you in the next steps of improvement and the final manuscript will be really valuable for the readers. This study demonstrates that thymopentin enhances antitumor immunity by promoting thymic rejuvenation, preventing T cell exhaustion, and synergizing with adoptive T cell therapy in murine cancer models. The findings highlight TP5 as a promising immunomodulatory agent that addresses key limitations of current T cell–based cancer immunotherapies.
Although the manuscript is well-prepared, some points still require polishing:
Comment 1: Mechanistic depth: While thymic rejuvenation and T cell functional restoration are shown, the molecular pathways mediating TP5’s effects remain insufficiently defined.
Response 1: We appreciate this important observation and acknowledge that our initial manuscript did not adequately elaborate on the molecular mechanisms underlying TP5's effects. In the revised manuscript, we have provided a comprehensive mechanistic framework for TP5's function in the discussion section (line 329-353), integrating our findings with existing literature on thymic biology and T cell immunology. However, we acknowledge that definitive elucidation of the precise molecular and cellular pathways remains an area requiring further investigation. As discussed in the study limitations section, future work will systematically dissect the molecular and cellular pathways underlying TP5-mediated thymic rejuvenation and T cell function preservation, including detailed analysis of signaling cascades, transcriptional programs, and metabolic reprogramming.
Comment 2: Translational gap: Evidence is limited to murine models. Human relevance and safety in cancer contexts remain speculative. Could you elaborate on this?
Response 2: Thank you for this critical point regarding clinical translation. We have collected clinical evidence that supports both the safety and efficacy of thymopentin in cancer patients. Here we summarized as below:
Clinical Evidence in NSCLC: A comprehensive systematic review and meta-analysis of 27 randomized controlled trials involving 1,925 NSCLC patients in China has provided robust evidence for synthetic thymic peptides (including thymopentin's) clinical efficacy and safety (PMID: 31326719). This meta-analysis demonstrated that combining thymopentin with chemotherapy significantly increased the objective response rate and disease control rate compared to chemotherapy alone. The 1-year overall survival rate was significantly improved. Combined treatment significantly enhanced CD3+ T cells, CD3+CD4+ T cells, natural killer cells, and the CD4+/CD8+ T cell ratio, indicating improved antitumor immunity.
Safety and tolerability: Combined treatment resulted in significantly lower risks of neutropenia and gastrointestinal reactions compared to chemotherapy alone.
Additional Clinical Studies: A clinical study in 60 NSCLC patients (PMID: 34714020) demonstrated that thymopentin combined with ribonucleic acid significantly improved treatment efficacy, reduced complications, and enhanced immune function (improving CD4+/CD8+ ratios and IL-2 activity) compared to ribonucleic acid alone, with excellent tolerability and no significant adverse events.
Furthermore, a prospective randomized trial in patients with advanced carcinoma undergoing highly cytotoxic chemotherapy (PMID: 7516641) showed that thymopentin was associated with a reduction in febrile episodes (52% versus 64% in placebo). Tolerance to thymopentin was excellent, with no significant adverse events reported. These clinical data collectively confirm TP5’s favourable safety profile in cancer patients and support its translational potential.
Comment 3: Overinterpretation: The discussion sometimes makes strong claims (e.g., “paradigm-shifting approach”) without direct clinical evidence, which may weaken credibility. Please, take a look on the discussion and provide strong evidence.
Response 3: We appreciate this insightful comment and have carefully reviewed the entire discussion section and moderated our language to ensure claims are appropriately supported by our data and existing literature. Specifically:
We have removed or qualified overly strong statements such as "paradigm-shifting approach" and replaced them with more measured language such as "promising therapeutic strategy" as described in line 392.
Comment 4: Visualisation: Would it be possible to add some figures and graphs too? It will definitely increase the value of your already well-written manuscript. There are several mechanisms and many molecules described in the paper, it may not be easy to digest for younger readers, and good pictures showing these pathways and molecules, also aberrant places, will definitely be of great value.
Response 4: We sincerely appreciate this excellent suggestion to enhance the visual presentation of our manuscript. As noted, our original manuscript already includes a comprehensive schematic diagram illustrating the mechanism by which TP5 promotes thymic rejuvenation and T cell functional restoration, as presented in Discussion. This figure depicts the key cellular processes involved in TP5's mechanism of action presented in our study.